# OpenReview forum: "On the Generalization of Dynamic GNNs: A Heavy-Tailed Wavelet Perspective"
_ICLR.cc/2026/Conference — ICLR 2026 Conference Withdrawn Submission_

### Official Review · Reviewer_rGAK · 2025-10-19

**Soundness:** 3
**Presentation:** 1
**Contribution:** 2
**Rating:** 2
**Confidence:** 3

**Summary:**

This paper focuses on the generalization problem of Dynamic Graph Neural Networks (DGNNs) in handling dynamic graphs with bursty and intermittent dynamics. The core innovation is the discovery that node-wise temporal signals, when transformed into the wavelet domain, exhibit Pareto-type heavy-tailed distributions, and the proposal of a Tail-Aware Masking mechanism that retains only the top wavelet coefficients (per node) and zeros out the rest before message passing. Theoretically, under the assumption of a regularly varying tail with index α>2, the paper proves that the retained coefficients capture a constant fraction of energy scaling, masking increases the effective tail index of features. Experimentally, on the METR-LA dataset, the tail-aware DGNN consistently outperforms the baseline, achieving  reduction in MSE and gains on tail-sensitive metrics while maintaining training stability through a short warmup period. However, the paper could still improved in clarity, comprehensiveness of experiments.

**Strengths:**

1. The paper has a clear research motivation, which originates from the discovery of the heavy-tailed distribution of wavelet coefficients. It subsequently constructs a theoretical framework and proceeds with the design of the corresponding method, forming a logical research process.

**Weaknesses:**

1. The clarity of the paper needs significant improvement:
   - Some abbreviations are used without prior full-form explanations, such as POT, Q-Q plot, and CCDF.
   - The specific formulation and application of the "DGNN" are not clearly elaborated, and details regarding model training and evaluation (e.g., representation learning strategies, loss function designs) are lacking.
   - The writing formality is insufficient: the pipeline in Section 4.4 only provides a brief procedure without detailed descriptions, and the experimental tasks mentioned in lines 313–323 are ambiguous due to the absence of necessary explanations.

2. The experimental comparisons lack comprehensiveness. Both recent and classic baseline models in the field of dynamic graph learning—especially wavelet-based models (e.g., [1,2,3])—are missing. Only one baseline model is adopted, which weakens the persuasiveness of the experimental results.

3. The paper fails to conduct a time complexity analysis, particularly regarding the wavelet decomposition process applied to dynamic graphs.



[1] Sun, Ke, et al. "ModWaveMLP: MLP-based mode decomposition and wavelet denoising model to defeat complex structures in traffic forecasting." Proceedings of the AAAI conference on artificial intelligence. Vol. 38. No. 8. 2024.

[2] Jiang, Jiawei, et al. "Pdformer: Propagation delay-aware dynamic long-range transformer for traffic flow prediction." Proceedings of the AAAI conference on artificial intelligence. Vol. 37. No. 4. 2023.

[3] Li, Yaguang, et al. "Diffusion convolutional recurrent neural network: Data-driven traffic forecasting." arXiv preprint arXiv:1707.01926 (2017).

[4] Chen, Chao, et al. "Easydgl: Encode, train and interpret for continuous-time dynamic graph learning." IEEE Transactions on Pattern Analysis and Machine Intelligence (2024).

**Questions:**

1. How to differentiate the nosie and useful information when using the masking?
2. As wavelet decoposition is very time-consuming with high computational cost, how does it performs in time complexity as well as running time?

---

### Official Review · Reviewer_LBoN · 2025-11-06

**Soundness:** 1
**Presentation:** 2
**Contribution:** 2
**Rating:** 2
**Confidence:** 3

**Summary:**

This paper examines dynamic graph node time-series through a wavelet lens and observes heavy-tailed Pareto-like distributions in their coefficients. It proposes a Tail-aware masking mechanism that retains only high-magnitude wavelet coefficients before message passing in a DGNN.

**Strengths:**

**1. Interesting conceptual framing**

The paper organizes existing wavelet- and heavy-tail-based intuitions into a unified viewpoint on temporal graph signals, though the underlying ideas are not new.

**2. Simple and interpretable mechanism**

The masking scheme is easy to integrate and interpretable.

**Weaknesses:**

**1. Major methodological flaw**

This paper repeatedly claim to study “dynamic graphs” with formulation of $\mathcal{G}_{t}$ (the graph topology is time-varing); yet the quantitative validation is done on METR-LA, whose graph topology does not change over time.

This leads to a major problem that the heavy-tailed phenomenon they observe is purely due to temporal signal bursts, not structural dynamics. Consequently, the title “Generalization of Dynamic GNNs” and much of the narrative are misaligned with the experiments.

**2. Method novelty is minor**

The Tail-Aware Masking is effectively quantile-based feature pruning, a trivial operation already present in sparse attention and magnitude-pruning literature. There is no comparison to existing sparsification techniques (attention masking, dropout variants, learned gating).

**3. The theorectical analysis lacks rigor and originality**

In Sec. 4.5 (Complexity and gneralisation), the derivation of the Rademacher-complexity bound is a sketch; actually, the claimed bound $\mathcal{O}(\rho^{1/2-1/\alpha}/\sqrt{nT})$ is just a restating of classical i.i.d. heavy-tail results, not a new theoretical contribution for DGNNs.

**4. Missing ablation**

This paper reports limited visual results. For example, in terms of Retention ratio $\rho$, the paper simply claims that ``Performance improves as $\rho$ decreases from 10'', but no numerical evidence is provided. Similarly, other hyperparameters (e.g., warmup length, Huber loss parameter $\delta$) are discussed descriptively but lack quantitative comparisons or sensitivity analyses.

**Questions:**

Please see weaknesses.

---

### Official Review · Reviewer_3vf2 · 2025-11-10

**Soundness:** 2
**Presentation:** 2
**Contribution:** 2
**Rating:** 4
**Confidence:** 2

**Summary:**

The paper analyzes dynamic graph signals (time series) using wavelet transforms and reports a heavy-tailed pattern in the coefficients.

**Strengths:**

The research identifies and formalizes the phenomenon of heavy-tailed distributions in the wavelet coefficients of dynamic graph signals. This discovery provides a previously overlooked perspective for understanding dynamic graph data.

**Weaknesses:**

The paper does not meet the standards of ICLR for the following reasons:

Writing and presentation: The paper is poorly organized and lacks self-containment. Many essential details are missing. For instance, Section 3 is overly concise, and the experimental setup for the reported findings is not described, making it difficult to assess their validity.

Theoretical formulation: The theory is not well developed. The underlying assumptions are unclear, and the role of “dynamic graphs” in the formulation is insufficiently explained.

Algorithmic contribution: The proposed algorithm, which is said to follow from the theoretical observations, is not presented clearly or in sufficient detail.

Overall, the submission appears incomplete and requires substantial improvement in both technical depth and presentation before it can be considered ready for publication.

**Questions:**

Please refer to the weaknesses.

---

### Note · Authors · 2025-11-17

I have read and agree with the venue's withdrawal policy on behalf of myself and my co-authors.